Prognostic value of high-sensitivity cardiac troponin for major adverse cardiovascular events in patients with diabetes: a systematic review and meta-analysis

Song Tiange
Lan Yu
Li Kecheng
Huang Honglang
Jiang Li u0341992@yahoo.com
1 Sichuan Provincial Key Laboratory for Human Disease Gene Study, University of Electronic Science and Technology of China , Chengdu , China
2 Department of Laboratory Medicine, Sichuan Provincial People’s Hospital , Chengdu , China
3 Research Unit for Blindness Prevention of Chinese Academy of Medical Sciences (2019RU026), Sichuan Academy of Medical Sciences , Chengdu , China
Uversky Vladimir
Electronic publication date: 2023 Nov 13
Publication date: 2023
Volume: 11
Electronic Location ID: e16376
Received 2023 Aug 8; Accepted 2023 Oct 8
Copyright: ©2023 Song et al.
Copyright year: 2023
Copyright holder: Song et al.
License: This is an open access article distributed under the terms of the Creative Commons Attribution License, which permits unrestricted use, distribution, reproduction and adaptation in any medium and for any purpose provided that it is properly attributed. For attribution, the original author(s), title, publication source (PeerJ) and either DOI or URL of the article must be cited.
License URL: https://creativecommons.org/licenses/by/4.0/

Keywords: High-sensitivity cardiac troponin, Major adverse cardiovascular event, Diabetes, Prognosis

Funding: The Natural Science Foundation of China 82121003 The Grant from Chinese Academy of Medical Sciences 2019-I2M-5-032 This work was supported by the Natural Science Foundation of China (No. 82121003), and the Grant from Chinese Academy of Medical Sciences (No. 2019-I2M-5-032). The funders had no role in study design, data collection and analysis, decision to publish, or preparation of the manuscript.

==============================
Background

High-sensitivity cardiac troponin (hs-cTn) is associated with cardiovascular outcomes in the general population, but the prognostic value of hs-cTn in the diabetic population remains inconclusive. This study aimed to systematically review current evidence regarding the association between hs-cTn and the prognosis of diabetic patients.

Methods

MEDLINE, Embase, and the Cochrane Database were searched from inception to May, 2023. Observational studies that investigated the prognostic value of hs-cTn in diabetic patients were included in this meta-analysis. Studies were excluded if they did not report outcomes of interest, or urine hs-cTn were measured. Two independent investigators extracted and analyzed the data according to the PRISMA guidelines. The primary outcome was long-term major adverse cardiovascular events (MACE).

Results

We included 30 cohort studies of 62,419 diabetic patients. After a median follow-up of 5 (4.1–9.5) years, the pooled results suggested elevation of hs-cTn was associated with a significantly increased risk of MACE (adjusted hazard ratio (HR) per standard deviation (SD) change 1.15, 95% CI [1.06–1.25], I2 = 0%) and heart failure (adjusted HR per SD change 1.33, 95% CI [1.08–1.63], I2 = 0%) in patients with diabetes. No significant association was found regarding the association between elevation of hs-cTn and risk of all-cause mortality (adjusted HR per SD change 1.24, 95% CI [0.98–1.57], I2 = 0%). The results of sensitivity analyses were similar in prospective cohort studies, high-quality studies, or population without major cardiovascular comorbidities at baseline. hs-cTn may represent a strong and independent predictor of MACE and heart failure in diabetic patients. Future research is warranted to determine the appropriate cutoff value for hs-cTn with different comorbidities, for instance, diabetic nephropathy, peripheral artery diseases, etc.

Introduction

Diabetes currently affects approximately 422 million people worldwide. Over the past two decades, the global prevalence among adults has nearly doubled, escalating from 4.7% to 8.5% (Chatterjee, Khunti & Davies, 2017; NCD Risk Factor Collaboration, 2016; DiMeglio, Evans-Molina & Oram, 2018). It was estimated that diabetes directly resulted in 1.5 million deaths annually. When considering the impact of higher-than-optimal blood glucose levels, an additional 2.2 million deaths are attributed (World Health Organization, 2016). Major adverse cardiovascular events (MACE) contributed to the majority of deaths related to high blood glucose (Pylypchuk et al., 2021). As the level of blood glucose rises, the risk of MACE increases continuously even before reaching the diagnostic standard of diabetes (Danaei et al., 2006). Patients diagnosed with diabetes can have over two-fold higher risk of cardiovascular disease compared to adults with normal blood glucose (Emerging Risk Factors et al., 2010). Considering the huge economic burden imposed by MACE related to diabetes (NCD Risk Factor Collaboration, 2016), predicting and identifying patients at high risk of MACE and related adverse events is of great clinical importance.

Recent evidence revealed that cardiac biomarkers play a crucial role in stratification for cardiovascular risk, and could add significant predictive increment compared to conventional models (Prugger et al., 2013; Wong et al., 2018). Among those widely studied biomarkers, high-sensitivity cardiac troponin (hs-cTn) has garnered notable attention in improving the prediction of cardiovascular outcomes (Willeit et al., 2017). A previous meta-analysis of 96,702 individuals suggested that high levels of hs-cTn was associated with a significantly increased risk of stroke in the general population during a median follow-up of ten years (Broersen et al., 2020). Furthermore, another meta-analysis of prospective studies indicated a strong association between hs-cTn and the risk of first-ever heart failure in asymptomatic subjects (Evans et al., 2018). However, the current predictive value of hs-cTn was mainly based on the general population, the role of hs-cTn in predicting MACE in diabetic patents is still controversial.

The presence of diabetes inherently elevates levels of cardiac biomarkers including hs-cTn (De Marco et al., 2021), thus predictive value of hs-cTn in cardiovascular outcomes may be sheltered to some degree. To date, there are increasing prospective data on the use of hs-cTn in risk stratification in diabetic patients, but the results are inconclusive (Busch et al., 2021; Pandey et al., 2021; Saeed et al., 2021; Witkowski et al., 2021). To quantitatively summarize current evidence, the aim of this comprehensive systematic review and meta-analysis was to evaluate the association between hs-cTn levels and risk of MACE as well as other adverse events in patients with diabetes or prediabetes.

Materials & Methods

Review protocol

This systematic review and meta-analysis was conducted according to the reporting guidelines of Meta-analysis of Observational Studies in Epidemiology (MOOSE) (Stroup et al., 2000) (checklist is shown in Appendix A). The protocol was published on the International prospective register of systematic reviews (PROSPERO) (CRD42021287365).

Data sources and searches

As for data sources, two authors (T.S. and Y.L.) searched MEDLINE, Embase, and the Cochrane Database for potential eligible original articles. The following keywords were used: “troponin T”, “troponin I”, “cardiac troponin”, “high-sensitivity”, “high-sensitive”, “diabetes mellitus”, “diabetes”. No language restriction was applied. We also manually screened the reference lists of included studies for potential articles meeting inclusion criteria. The electronic literature search was last updated on May 25th, 2023. The detailed search strategies are displayed in Appendix B.

Study selection

Two authors (T.S. and Y.L.) independently selected eligible studies based on standard inclusion and exclusion criteria. Disagreements were resolved by discussion with a third author (LJ). In this systematic review, observational cohort studies were included if they investigated the association between hs-cTn and risk of cardiovascular outcomes in patients with diabetes or prediabetes. Studies were excluded if they met the following standards: (1) the study did not report data regarding cardiovascular outcomes or mortality, (2) urine hs-cTn were measured instead of plasma hs-cTn.

Data extraction

Data collection was accomplished by two authors using a unified extraction form independently, and any discrepancy was handled by discussion from a third author (LJ). If one article contained data for different populations, for instance, prediabetes and diabetes, it was then extracted as separate studies.

The primary outcome was MACE, which was defined as the composition of incident cardiovascular events, and hospitalization or mortality with a primary or secondary diagnosis of cardiovascular diseases. Data of outcomes which involved definitions of incident cardiovascular diseases, or ischemic heart disease, or cardiovascular mortality were also extracted as MACE. The secondary outcomes included all-cause mortality, and incident heart failure.

Quality assessment

Two investigators independently assessed the risk of bias of included studies with Newcastle-Ottawa Scale (NOS) (Wells, O’connell & Peterson, 2000). The eligible cohort studies were scored on the following domains: representativeness of the study population, selection of non-exposed cohort population, assays of measuring hs-cTn, demonstration that outcomes of interest was not present at the beginning of study, comparability of cohort with appropriate adjustment for confounding factors, assessment of outcomes, adequate length of follow-up and loss to follow-up. Each included study was allocated stars based on the above domains (from zero to nine), and studies with a higher score than seven were determined to be high-quality.

Data synthesis and analysis

As our meta-analysis focused on long-term outcomes involving time-to-event data, hazard ratio (HR) and related 95% confidence interval (95% CI) were used to summarize the effect measures according to the Cochrane Handbook for Systematic Reviews of Interventions (version 6.2) (Higgins, Li & Deeks, 2021). The values of HR and 95% CI were obtained from Cox proportional hazards regression analyses on a continuous scale (per standard deviation (SD) change after log transformation) or a categorical scale (high versus low levels based on included studies). Generic inverse-variance methods were used to pool the results of the natural log transformed HR (lnHR) and related standard errors (SE (lnHR)) (Deeks, Higgins & Altman, 2021). As innate and unmeasurable heterogeneity may exist in study population and hs-cTn measurement assays across included studies, we adopted random-effect model to pool the results of each study (Deeks, Higgins & Altman, 2021).

The chi-squared (χ2) test and I2 statistic were used to assess the inter-study heterogeneity (Higgins et al., 2003). The rough interpretation of the I2 statistic was as follows: 0% to 40%, might not be important; 30% to 60%: moderate heterogeneity; 50% to 90%: substantial heterogeneity; 75% to 100%: considerable heterogeneity. Subgroup analyses were performed to address potential innate heterogeneity, stratified by types of diabetes and subunit of hs-cTn. In addition, several sensitivity analyses were performed to evaluate the robustness of main analyses: (1) only including perspective studies, as prospective studies usually have fewer potential sources of bias and confounding than retrospective studies; (2) only including high-quality studies derived from quality assessment; (3) only including studies adequately adjusted for demographics, cardiovascular risk factors, biomarker levels or medication use, etc.; (4) only including the population without major cardiovascular comorbidities at the start of study. In addition, Funnel plots and the Egger test were used to assess the potential for publication bias. Review Manager V.5.1 (The Nordic Cochrane Centre, København, Denmark) was used for statistical analysis.

Figure 1 Flow diagram of study selection.

Results

Study characteristics

After screening 601 records retrieved from the systematic literature search, we finally identified 26 articles eligible for our study (Bidadkosh et al., 2017; Bluro et al., 2021; Busch et al., 2021; Cimaglia et al., 2021; Colombo et al., 2018; Costacou, Saenger, & Orchard, 2020; Galsgaard et al., 2017; Gori et al., 2016; Hendriks et al., 2016; Hillis et al., 2014; Junttila, 2016; Keller et al., 2018; Looker et al., 2015; Nguyen et al., 2020; Ohkuma et al., 2017; Pandey et al., 2021; Price et al., 2017; Resl et al., 2016; Saeed et al., 2021; Scirica et al., 2016; Sharma et al., 2020; Tang et al., 2020; Witkowski et al., 2021; Wong et al., 2019; Yiu et al., 2014; Zellweger et al., 2015). Four articles involved different cohorts, thus they were treated as separate studies (Galsgaard et al., 2017; Nguyen et al., 2020; Pandey et al., 2021; Tang et al., 2020). Therefore, a total of 30 studies were included in the quantitative analysis of our study. The detailed study selection flow diagram is shown in Fig. 1. Among the included studies, 29 were prospective cohort studies, one was a retrospective cohort study. Most studies measured the subunit of high-sensitivity cardiac troponin T (hs-cTnT), with only three examining high-sensitivity cardiac troponin I (hs-cTnI). Sixteen studies investigated population with type 2 diabetes, among which two studies included populations with prior stable coronary artery diseases or recent acute coronary syndrome, two studies with chronic kidney diseases. Four studies focused on patients with type 1 diabetes, and three on patients with prediabetes. In total, 62,419 participants were included. The median follow-up time of involved population was 5 (4.1–9.5) years. The detailed characteristics of involved studies are shown in Table 1.

Table 1 Baseline characteristics of the included studies.

Author	Year	Study design	Subunit of hs-cTn (Assay)	Median hs-cTn in ng/La	Population	HbA1c (%)a	Sample size	Duration of follow-up (years)a	
Witkowski et al.	2021	prospective	hs-cTnT (Roche Diagnostics)	13 (8.2–21.6)	Prediabetes	5.8 (5.5–6.0)	2,631	5 years	
Saeed et al.	2021	retrospective	hs-cTnT (Roche Diagnostics)	4.0 ± 4.8	T1DM	8.2 ± 1.2	295	14.4 (0.5–16) years	
Pandey et al.	2021	prospective	hs-cTnT (Roche Diagnostics)	3.0 (1.5–6.0)	Prediabetes	NA	4,543	10 years	
Pandey et al.	2021	prospective	hs-cTnT (Roche Diagnostics)	5.0 (1.5–8.9)	Diabetes	NA	2,256	10 years	
Cimaglia et al.	2021	prospective	hs-cTnT (Roche Diagnostics)	31 (20–59)	Diabetes with CLI	7.3 (6.4–8.4)	618	981 (557–1325) days	
Busch et al.	2021	prospective	hs-cTnI (Snibe Diagnostics)	2.4 (1.0–5.1)/2.6 (1.0–5.6)	T2DM	7.4 ± 1.4/7.7 ± 1.7	1,030	4.7 (4.0–5.3) years	
Bluro et al.	2021	prospective	hs-cTnT (Roche Diagnostics)	9 (6–13)	T2DM	7.8 (7.1–9.1)	482	2.5 years	
Tang et al.	2020	prospective	hs-cTnI (Abbott Diagnostics)	2.8 (1.9–4.5)/4.3 (2.8–7.7)	Diabetes	6.5 ± 1.0/6.7 ± 1.1	1,835	6.2 years	
Tang et al.	2020	prospective	hs-cTnT (Roche Diagnostics)	10.0 (7.0–14.0)/14.0 (9.0–22.0)	Diabetes	6.5 ± 1.0/6.7 ± 1.1	1,835	6.2 years	
Sharma et al.	2020	prospective	hs-cTnI (Abbott Diagnostics)	9	T2DM with recent ACS	NA	5,154	18 months	
Nguyen et al.	2020	prospective	hs-cTnT (Roche Diagnostics)	7.1 ± 7.8	Prediabetes	NA	799	12.4 ± 3.8 years	
Nguyen et al.	2020	prospective	hs-cTnT (Roche Diagnostics)	10.5 ± 16.2	Diabetes	NA	695	12.4 ± 3.8 years	
Costacou, Saenger & Orchard	2020	prospective	hs-cTnT (Roche Diagnostics)	5.0 (<3.0–10.0)	T1DM	8.7 (7.9–9.8)	581	20.3 years	
Wong et al.	2019	prospective	hs-cTnI (Abbott Diagnostics)	2.2 ± 1.1	T2DM with stable CAD	7.38 ± 1.36	1,617	51 months	
Keller et al.	2018	prospective	hs-cTnT (Roche Diagnostics)	55 (35-90)	T2DM with hemodialysis	6.74 ± 1.22	1,034	4.04 (3.89–4.20) years	
Colombo et al.	2018	prospective	hs-cTnT (Glasgow Biomarker Laboratory)	1.5 (1.5–5.5)/4.1 (1.5–9.1)	T2DM	7.7 (6.8–8.7)/7.6 (6.9–8.4)	2,105	4.1 (3.3–4.7) years	
Price et al.	2017	prospective	hs-cTnT (Roche Diagnostics)	9.6 (6.9–13.8)	T2DM	NA	1,049	8 years	
Ohkuma et al.	2017	prospective	hs-cTnT (Roche Diagnostics)	5.0 (1.5–10.0)	T2DM	7.4 ± 1.4	3,098	5 years	
Galsgaard et al.	2017	prospective	hs-cTnT (Roche Diagnostics)	3.14 (1.1–6.0)	T1DM without nephropathy	8.4 ± 1.1	442	8.1 (6.6–12.6) years	
Galsgaard et al.	2017	prospective	hs-cTnT (Roche Diagnostics)	8.9 (4.1–17.2)	T1DM with nephropathy	9.4 ± 1.5	458	8.1 (6.6–12.6) years	
Bidadkosh et al.	2017	prospective	hs-cTnT (Roche Diagnostics)	30 (20–47)	T2DM with nephropathy	8.0 ± 1.6	861	9 (4–17) months	
Scirica et al.	2016	prospective	hs-cTnT (Roche Diagnostics)	12.0 (8.1–18.4)	T2DM	7.6 (6.9, 8.7)	16,492	2.1 (1.8–2.3) years	
Resl et al.	2016	prospective	hs-cTnT (Roche Diagnostics)	0.0008 (0.005–0.013)	T2DM	7.3 ± 1.1/7.1 ± 1.2	746	60 (60–60) months	
Junttila	2016	prospective	hs-cTnT (NA)	NA	T2DM	NA	2,285	5 years	
Hendriks et al.	2016	prospective	hs-cTnT (NA)	NA	T2DM	7.2 ± 1.3	1,133	11 (7–14) years	
Gori et al.	2016	prospective	hs-cTnT (Roche Diagnostics)	6 (3–10)	Diabetes	NA	1,510	13.1 (6.9–14.4) years	
Looker et al.	2015	prospective	hs-TnT(RBM)	NA	T2DM	6.9 (6.5–7.5)/7.0 (6.6–7.8)	2,318	3.2(1.5–4.9)/6.5(3.9–7.9) years	
Zellweger et al.	2015	prospective	hs-cTnT (Roche Diagnostics)	9.0 (5.0–22.4)	Diabetes with suspected AMI	NA	379	814 days	
Hillis et al.	2014	prospective	hs-cTnT (Roche Diagnostics)	5.0 (1.5–11.0)	T2DM	7.41 ± 1.43	3,862	5 years	
Yiu et al.	2014	prospective	hs-TnI (Abbott Diagnostics)	4.8 (3.2–8.4)	T2DM	7.8 ± 1.4	276	4.9 (3.7–5.6) years	
Notes.

Abbreviations hs-cTn high-sensitivity cardiac troponin

HR hazard ratio

SD standard deviation

OR odds ratio

IQR interquartile range

T1DM type 1 diabetes mellitus

T2DM type 2 diabetes mellitus

CLI critical limb ischemia

ACS acute coronary syndrome

CAD coronary artery disease

AMI acute myocardial infarction

HbA1c Hemoglobin A1C

NA not available

a Data was presented as mean ± standard deviation or median (interquartile range).

Quality assessment

The median NOS score for included studies was 8, with 17 studies scored of 8, eight studies of 7, and five studies of 6. As for selection of study population, all included studies had optimal representatives of exposed cohort and selection of non-exposed cohort, as well as valid ascertainment of exposure, but ten studies partially contained patients with prior cardiovascular diseases. Most included studies (24/30) had adequate adjustment for potential confounding factors, involving demographics, comorbidities, biomarker levels, medications, etc. Detailed adjusted variables were shown in Table S1. Nearly all studies had a median follow-up period of over two years, but few of them mentioned the percentage of patients lost during follow-up. The summary of quality assessment was displayed in Fig. 2.

Figure 2 Risk of bias of included studies.

Scoring was based on Newcastle-Ottawa Scale. Included studies: Witkowski et al. (2021); Saeed et al. (2021); Pandey et al. (2021); Cimaglia et al. (2021); Busch et al. (2021); Bluro et al. (2021); Tang et al. (2020); Sharma et al. (2020); Nguyen et al. (2020); Costacou, Saenger & Orchard (2020); Wong et al. (2019); Keller et al. (2018); Colombo et al. (2018); Price et al. (2017); Ohkuma et al. (2017); Galsgaard et al. (2017); Bidadkosh et al. (2017); Scirica et al. (2016); Resl et al. (2016); Junttila (2016); Hendriks et al. (2016); Gori et al. (2016); Looker et al. (2015); Zellweger et al. (2015); Hillis et al. (2014); and Yiu et al. (2014).

MACE

A total of 24 studies including 47,612 patients investigated the association between hs-cTn and risk of MACE. Eighteen studies evaluated the prognostic effect of hs-cTn from a continuous scale, and the pooled results suggested that elevation of hs-cTn was associated with a significantly increased risk of MACE in patients with diabetes (adjusted HR per standard deviation (SD) change 1.15, 95% CI [1.06–1.25], Fig. 3). No heterogeneity was observed among included studies (I2 = 0%). When dividing the hs-cTn levels into high and low categories, we also found significant association between high hs-cTn and increased risk of MACE without heterogeneity (adjusted HR 1.73, 95% CI [1.11–2.68], I2 = 0%). Most studies adopted 14 ng/L as the cutoff value of hs-cTn. No significant publication bias was detected in the Egger test (t = 0.52, p = 0.62) and funnel plot (Fig. S1).

Figure 3 Forest plot regarding hs-cTn and risk of MACE in diabetic patients.

SE, standard error; IV, inverse-variance methods; hs-cTn, high-sensitivity cardiac troponin; MACE, major adverse cardiovascular event.

When stratified by types of diabetes, the pooled results of 12 studies suggested that elevation of hs-cTn was associated with a significantly increased risk of MACE in type 2 diabetes (adjusted HR per SD change 1.14, 95% CI [1.04–1.25], I2 = 0%), while no significant association was found between elevation of hs-cTn and risk of MACE in type 1 diabetes (adjusted HR per SD change 1.17, 95% CI [0.93–1.47], I2 = 0%). As for the subunit of hs-cTn, we observed significant association of increased risk of MACE with hs-cTnT (adjusted HR per SD change 1.19, 95% CI [1.08–1.31], I2 = 0%), but not with hs-cTnI (adjusted HR per SD change 1.06, 95% CI [0.91–1.24], I2 = 0%). The results of sensitivity analyses were similar in perspective studies, high-quality studies, adequately adjusted studies, or population without major cardiovascular comorbidities at baseline (Table 2).

Table 2 Summary of pooled results of overall and sensitivity analyses.

	MACE	Heart failure	All-cause mortality	
	adjusted HR	95% CI	adjusted HR	95% CI	adjusted HR	95% CI	
Overall analyses							
per SD change	1.15	1.06–1.25	1.33	1.08–1.63	1.24	0.98–1.57	
high vs low	1.73	1.11–2.68	1.59	0.91–2.78	1.69	0.93–3.10	
Sensitivity analyses							
prospective studies	1.15	1.06–1.25	1.33	1.08–1.63	1.24	0.98–1.57	
high-quality studies	1.26	1.07–1.49	1.33	1.03–1.72	1.56	0.82–2.96	
adequately adjusted studies	1.20	1.08–1.32	1.33	1.08–1.63	1.56	0.82–2.96	
population without cardiovascular events at baseline	1.14	1.02–1.28	1.33	1.03–1.72	1.68	0.79–3.60	
Notes.

MACE major adverse cardiovascular events

HR hazard ratio

SD standard deviation

CI confidence interval

Heart failure

The pooled results of ten studies involving 28,159 diabetic patients suggested elevation of hs-cTn was associated with a significantly increased risk of heart failure (adjusted HR per SD change 1.33, 95% CI [1.08–1.63], I2 = 0%, Fig. 4). Similar results were found in the multiple sensitivity analyses (Table 2). But after dividing the concentrations of hs-cTn with a cutoff value of about 6 ng/L, no significant association was observed between high hs-cTn and risk of heart failure (adjusted HR 1.59, 95% CI [0.91–2.78], I2 = 0%).

Figure 4 Forest plot regarding hs-cTn and risk of heart failure in diabetic patients.

SE, standard error; IV, inverse-variance methods; hs-cTn, high-sensitivity cardiac troponin.

In the subgroup analyses of prediabetes and diabetes population, the pooled results indicated elevation of hs-cTn was associated with a significantly increased risk of heart failure in patients with diabetes (adjusted HR per SD change 1.39, 95% CI [1.09–1.77], I2 = 0%), but not in patients with prediabetes (adjusted HR per SD change 1.19, 95% CI [0.82–1.73], I2 = 0%).

All-cause mortality

Eleven studies with 12,874 patients evaluated the association between hs-cTn and risk of all-cause mortality in diabetic population. The pooled results from four studies showed no significant association between elevation of hs-cTn and risk of all-cause mortality (adjusted HR per SD change 1.24, 95% CI [0.98–1.57], I2 = 0%). The results remained similar after dividing the level of hs-cTn into high and low categories (adjusted HR 1.69, 95% CI [0.93–3.10], I2 = 0%, Fig. S2). Four studies adopted a cutoff value of hs-cTn ranging from 8.2 to 14 ng/L, and the rest of the studies used 25 or 35 ng/L.

Discussion

The results of this systematic review and meta-analysis further expanded the prognostic value of hs-cTn for future MACE into the population with diabetes. In a population of 62,419 patients from 30 cohorts, we found elevation of hs-cTn was associated with a significantly increased risk of MACE, heart failure, but not all-cause mortality in patients diagnosed with diabetes. After adopting a cutoff value of 14 ng/L, high level of hs-cTnT was also associated with a significantly increased risk of MACE. Notably, the prognostic utility of hs-cTn was more pronounced in patients with type 2 diabetes compared to those with type 1 diabetes or prediabetes. Additionally, hs-cTnT appeared to offer superior prognostic insights than hs-cTnI.

Cardiac troponin was a well-established cardiac biomarker for predicting MACE in patients who underwent noncardiac or vascular surgeries (Borg Caruana et al., 2020; Zhang et al., 2018). However, the baseline levels of troponin may increase in certain populations, for instance, patients with diabetes or chronic kidney diseases, thus the diagnostic or prognostic value of troponin may be biased and limited (Yang et al., 2020). With the advent and development of hs-cTn assays, it’s now feasible to detect extremely low levels of hs-cTn. Consequently, subtle changes in hs-cTn might offer enhanced predictive insights for certain populations. Recent meta-analyses indicated that hs-cTn can act as a robust and independent predictor of MACE in patients with chronic heart failure (Aimo et al., 2018). However, inconclusive evidence limited the application of hs-cTn in patients with diabetes. It was noteworthy that the majority of the included studies suggested nonsignificant association between hs-cTn and risk of MACE, but the pooled results revealed that per unit elevation of hs-cTn was associated with a significantly increased risk of MACE.

Regarding the optimal cutoff value of hs-cTnT for MACE, most original studies adopted 14 ng/L, which lies in the range of 99th percentile of healthy subjects (10–20 ng/L) (Collet et al., 2021). This cutoff aligns with the value reported in previous individual patient data meta-analysis regarding patients with chronic heart failure (Aimo et al., 2018). These findings may be explained by the fact that even mild sustained myocardial damage can gradually accumulate and culminate in adverse left ventricle remodeling and poor cardiovascular outcomes (Motiwala et al., 2015). For diabetic patients with other complications (critical limb ischemia, chronic kidney diseases, etc.), a higher cutoff value was adopted, ranging from 25 to 55 ng/L. This difference may be related to the relatively higher level of median baseline hs-cTn (31 ng/L for diabetic patients with critical limb ischemia, and 55 ng/L for diabetic patients with hemodialysis). In general, a cutoff value of 14 ng/L may be considered to aid risk stratification of MACE in clinical practice, but this cutoff value should be optimally modified according to the comorbidities.

Discrepant results from subgroup analyses implied that the hs-cTnT subunit assay may have better predictive value for MACE in diabetic patients. Likewise, the superiority of hs-cTnT was also noted in Akershus cardiac examination (ACE) 3 study, which suggested stronger association with outcome of hs-cTnT compared with hs-cTnI in patients with suspected unstable angina pectoris (Tveit et al., 2020). However, as is pointed out in a large national genome-wide association study of 19,501 individuals, elevation of hs-cTnT was more strongly associated with increased risk of non-cardiovascular death, whereas hs-cTnI was more likely to associated with cardiovascular mortality (Welsh et al., 2019). The difference prognostic performance of T and I subunit in our study may be attributed to the relatively small numbers of studies measuring hs-cTnI in diabetic population. Future prospective studies are needed to better understand the prognostic value of hs-cTnI in diabetic patients..

We also noted an interesting finding that hs-cTn seemed to perform better in type 2 diabetes compared to type 1 diabetes. The potential reason may rely on the fact that the patients of type 1 diabetes were much younger than those with type 2 diabetes, with fewer cardiovascular risk factors. Given the concept that hsTn level is not elevated until vessel damage is present, the real prognostic difference of cTn in type 1 vs type 2 diabetes should adjust the condition of macro- and microvascular diseases. Besides, only four cohorts from three studies reported the role of cTn in type 1 diabetes, its prognostic value may be also limited by the sample size. More studies are needed to further explore the role of hs-cTn in patients with type 1 diabetes.

The present systematic review and meta-analysis had several strengths. First, nearly all included studies are of a prospective nature, reducing the potential for recall and misclassification biases inherent in retrospective designs. Second, our study encompasses a substantial population of 62,419 patients from 30 cohorts, ensuring strong statistical power of the pooled results. Third, the majority of the included studies performed adequate adjustment of confounding factors, and the consistency between sensitivity analyses of adjusted results and overall analyses further confirmed the independent prognostic value of hs-cTn after excluding the effect of confounding factors. Lastly, the inter-study heterogeneity of all outcomes was remarkably low, with a I2 statistic of 0%, which indicated that our results were stable and applicable to a wide population.

Despite the above strengths, our meta-analysis also had several limitations. First, partial included population from some studies had cardiovascular comorbidities at the start of the study. To address this issue, we performed a sensitivity analysis by excluding those studies, and the consistency implied the robustness of our results. Second, the cutoff values of hs-cTn among included studies were different, which may introduce bias when pooling the results on a categorical scale. In spite of this, no obvious heterogeneity was detected among studies. More prospective studies investigating the appropriate and united cutoff values of hs-cTn in diabetic patients are needed in the future. Third, only a few studies measured levels of hs-cTnI, thus the nonsignificant association between hs-cTnI and cardiovascular outcomes may be masked by the small sample size. The prognostic value of hs-cTnI in diabetic patients also needs to be further explored in future researches, especially regarding cardiovascular-specific outcomes and general clinical outcomes.

Conclusions

In the present systematic review and meta-analysis, hs-cTn may represent a strong and independent predictor of future MACE and heart failure in patients diagnosed with diabetes, over a long follow-up period. A cutoff value of 14 ng/L may be adopted in risk stratification for general diabetic population, but limited evidence exists regarding the optimal cutoff point for diabetic patients with severe complications, involving critical limb ischemia or chronic kidney diseases. Future researches are warranted to determine the appropriate cutoff value for hs-cTn in diabetic population with different comorbidities, for instance, diabetic nephropathy, peripheral artery diseases, etc.

Supplemental Information

Supplemental Information 1 Supplementary information of included studies

hs-cTn =high-sensitivity cardiac troponin, HR =hazard ratio, SD =standard deviation, OR =odds ratio, IQR =interquartile range, MACE =major adverse cardiovascular events, IHD =ischemic heart diseases, CVD =cardiovascular diseases, CAD =coronary artery disease, NA =not available.

a, data was presented as mean ±standard deviation or median (interquartile range).

Click here for additional data file.

Supplemental Information 2 Funnel plot of pooled results regarding hs-cTn and risk of MACE

Click here for additional data file.

Supplemental Information 3 Forest plot indicating the effect of hs-cTn on all-cause mortality

Click here for additional data file.

Supplemental Information 4 Raw data of the extraction form

Click here for additional data file.

Supplemental Information 5 Raw data extracted from original article for Figs. 3 and 4

Click here for additional data file.

Supplemental Information 6 PRISMA checklist

Click here for additional data file.

Supplemental Information 7 Rationale and contribution of this study

Click here for additional data file.

Supplemental Information 8 Newcastle-Ottawa quality assessment scale for observational studies

We did not have categorical data that are recorded numerically in our extraction form. For those columns recorded with numbers, they are NOS scores in each area, we uploaded the details of how to score NOS in the supplements

Click here for additional data file.

Supplemental Information 9 Appendix

Literature search details

Click here for additional data file.

Additional Information and Declarations

Competing Interests

Author Contributions

Data Availability

The authors declare there are no competing interests.

Tiange Song conceived and designed the experiments, performed the experiments, analyzed the data, prepared figures and/or tables, authored or reviewed drafts of the article, and approved the final draft.

Yu Lan analyzed the data, prepared figures and/or tables, and approved the final draft.

Kecheng Li performed the experiments, authored or reviewed drafts of the article, and approved the final draft.

Honglang Huang performed the experiments, analyzed the data, authored or reviewed drafts of the article, and approved the final draft.

Li Jiang conceived and designed the experiments, authored or reviewed drafts of the article, and approved the final draft.

The following information was supplied regarding data availability:

The raw data mainly contains data extraction form of original data from included studies of meta-analysis

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
