# Peer review of "Prognostic value of high-sensitivity cardiac troponin for major adverse cardiovascular events in patients with diabetes: a systematic review and meta-analysis"

_PeerJ, doi:10.7717/peerj.16376_

## Round 0.1 · original submission · Major Revisions

Please address the concerns of both reviewers and amend the manuscript accordingly.

**Language Note:** The review process has identified that the English language must be improved. PeerJ can provide language editing services - please contact us at copyediting@peerj.com for pricing (be sure to provide your manuscript number and title). Alternatively, you should make your own arrangements to improve the language quality and provide details in your response letter. – PeerJ Staff

Reviewer 1 ·

Basic reporting

I want to thank the authors for the opportunity to review this review and meta-analysis. The use of cTn as a prognostic marker is intriguing, and this review adds a little more insight into this biomarker's use. The language is clear and easy to read.

Line 61: the correct term is high sensitivity and not hypersensitivity.

Experimental design

The authors provide a clear and well-defined research question. The search strategy is adequate, and the selection of studies seems correct. It is earlier reported that TnT performs slightly better than TnI as a prognostic marker. It is not clear why the authors focus on both cTnI and cTnT instead of focusing on just cTnT as it is only 3 studies using cTnI.

Line 107: It is not clear to me what this means. What do you mean by similar definitions?

Validity of the findings

It is really interesting why cTn performs better in Type 2 DM vs Type 1. Why do you think this is? Where there any correlations between Hba1c and cTn? Was there any data on time since diagnosis (how long have they been sick?)
I’m not convinced that you can conclude that a cut-off value of 14 ng/L could be used as a risk stratification tool based on your findings. Is not this just derived from the 99th percentile?
Some of your included studies involve patients with chronic kidney disease. Did you investigate how chronic kidney disease affected your analysis?

Reviewer 2 ·

Basic reporting

Abstract:

“Observational studies that investigated the prognostic value of hs-cTn in diabetic patients.” ---- No verb? Did you mean these studies were included? If so, why restricting to “observational studies”?

Recommend elaborating a bit more on the inclusion/exclusion criteria in the methods section of the abstract, if possible.

Where is the “median follow-up of 5 years” coming from? Median across the 30 included studies?

The authors mentioned “prospective cohort studies” in the results section but specified “observational studies” in the methods section? Why “observational studies” in the base case and yet “observational studies” in sensitivity analyses? Readers might benefit from more details or clearer description.

It is a bit surprising that the I-squared is as low as 0%.

Recommend being clear about the unit in change of hs-cTn to help readers with the interpretation of results (e.g., HR per SD change). Also, what is the full name of “SD”? This might be important, especially when the abstract specifies “future research is warranted to determine the appropriate cutoff value ofr hs-cTn with different comorbidities”. Also, the abstract didn’t specify other comorbidities. Suggest being a bit clearer.



Main text:

The comments above for the abstract are also applicable for the main text.

It would be nice to present the summary results of the quality assessment.

Line 127: “As substantial variation may exist in study population and hs-cTn measurement assays, we adopted random-effect model to pool the results of each study.” --- Recommend presenting evidence or analyses to justify the use of random-effect analysis without any fixed-effect analysis. The next sentence did specify the use of chi-squared and I-squared but recommend assessing the applicability of random-effect first before jumping into the choice of methods.

Line 274: “no obvious heterogeneity was detected among studies.” It seems to be contradictory to the prior statement “As substantial variation may exist in study population and hs-cTn measurement assays, we adopted random-effect model to pool the results of each study.”

Line 269: For the limitation section in general, recommend elaborating more on actions taken to mitigate these limitations (if any). It would be helpful for future researchers.

Experimental design

No comment

Validity of the findings

No comment

Additional comments

No comment

---

## Round 0.2 · accepted · Accept

All critiques were addressed and the revised manuscript is acceptable now.

Reviewer 1 ·

Basic reporting

The authors have provided me with sufficient answers to my questions.

Experimental design

No additional comments

Validity of the findings

No additional comments

Additional comments

No additional comments

Reviewer 2 ·

Basic reporting

The reviewers’ comments were addressed in a clear and organized manner. No further comment.

Experimental design

The reviewers’ comments were addressed in a clear and organized manner. No further comment.

Validity of the findings

The reviewers’ comments were addressed in a clear and organized manner. No further comment.